# Gradient SERS Substrates with Multiple Resonances for Analyte Screening: Fabrication and SERS Applications

**DOI:** 10.3390/molecules27165097

**Published:** 2022-08-10

**Authors:** Ashutosh Mukherjee, Quan Liu, Frank Wackenhut, Fang Dai, Monika Fleischer, Pierre-Michel Adam, Alfred J. Meixner, Marc Brecht

**Affiliations:** 1Center for Process Analysis and Technology (PA&T), School of Applied Chemistry, Reutlingen University, 72762 Reutlingen, Germany; 2Reutlingen Research Institute (RRI), Reutlingen University, 72762 Reutlingen, Germany; 3Institute of Physical and Theoretical Chemistry, Eberhard Karls University of Tübingen, 72076 Tübingen, Germany; 4Laboratory Light, Nanomaterials & Nanotechnologies–L2n and CNRS EMR 7004, University of Technology of Troyes, 10000 Troyes, France; 5Institute for Applied Physics, Eberhard Karls University of Tubingen, 72076 Tübingen, Germany; 6Center for Light-Matter-Interaction, Sensors and Analytics (LISA+), Eberhard Karls University of Tübingen, 72076 Tübingen, Germany

**Keywords:** surface-enhanced Raman spectroscopy (SERS), SERS substrates, multiple plasmonic resonances, island film

## Abstract

Surface-enhanced Raman spectroscopy (SERS) provides a strong enhancement to an inherently weak Raman signal, which strongly depends on the material, design, and fabrication of the substrate. Here, we present a facile method of fabricating a non-uniform SERS substrate based on an annealed thin gold (Au) film that offers multiple resonances and gap sizes within the same sample. It is not only chemically stable, but also shows reproducible trends in terms of geometry and plasmonic response. Scanning electron microscopy (SEM) reveals particle-like and island-like morphology with different gap sizes at different lateral positions of the substrate. Extinction spectra show that the plasmonic resonance of the nanoparticles/metal islands can be continuously tuned across the substrate. We observed that for the analytes 1,2-bis(4-pyridyl) ethylene (BPE) and methylene blue (MB), the maximum SERS enhancement is achieved at different lateral positions, and the shape of the extinction spectra allows for the correlation of SERS enhancement with surface morphology. Such non-uniform SERS substrates with multiple nanoparticle sizes, shapes, and interparticle distances can be used for fast screening of analytes due to the lateral variation of the resonances within the same sample.

## 1. Introduction

Since its discovery, Raman spectroscopy has attracted a lot of attention due to its selective molecular identification [1,2]. A major limitation is the inherently small Raman cross-section, which can be overcome with surface-enhanced Raman spectroscopy (SERS) [3]. On average, the Raman signal can be enhanced by a factor of 10^4^–10^6^, and sometimes even by a factor of 10^8^ by exploiting surface enhancement effects [4,5,6,7]. However, the challenge to achieve such high enhancements is strongly dependent on the design and fabrication of the SERS substrates. In many cases, a SERS substrate consists of noble metal nanoparticles (NPs), where the actual shape and distance of the NPs play a crucial role in SERS enhancement. The gap between the NPs has an especially significant impact on the SERS effect, since the electric field can be strongly enhanced in the gap due to confinement by the NPs [8,9,10,11,12]. Hence, the ideal SERS substrate needs to offer high Raman enhancement, which can be altered by manipulating the size, shape, and interparticle distance of noble metal NPs, and good stability (physically and chemically) to withstand any analyte. Furthermore, the fabrication process should be reproducible, simple, and cost-effective.

There are plenty of SERS substrates available on the market, such as surface-roughened films [13,14], colloidal NPs [15,16,17], metal/metal island films [18,19,20,21], etc. However, most of these substrates lack a certain combination of the aforementioned properties. A preferable substrate would be one that offers resonant excitation of plasmons in NPs and is stable, reproducible in geometry and plasmonic response, easy to prepare, and cost-effective. Plenty of SERS substrates have been fabricated with the aim of an optimal combination of the above-mentioned properties.

Many investigated substrates rely on the uniformity of NP sizes and shapes [22,23,24,25,26,27]. This uniformity yields a uniform Raman enhancement profile, which allows reproducible SERS results. However, for rapid and extensive screening to find the optimal combination of properties for SERS, it is desirable to have a SERS substrate that offers multiple NP sizes, shapes, and gaps leading to variable resonances.

Gradient assemblies of NPs with gradually changing sizes and densities have attracted interest because of their varied applications [28,29,30,31,32,33,34,35,36]. These samples fulfill many of the criteria mentioned above. Due to the gradual variation in the size of the NPs, their plasmonic response varies. These assemblies allow manipulation and investigation of their material properties at the micro and nano scales. Several techniques have have been employed to fabricate gradient NP assemblies on various substrates [28,32,36,37,38]. However, the definite distribution of sizes, shapes, and interparticle distances between NPs can hardly be controlled, especially in micro and nanometer scale gradients.

In this work, we report a facile method of fabricating non-uniform SERS substrates that not only have typical SERS substrate characteristics, but also inhomogeneity in NP sizes and gaps. We show that such a substrate offers multiple plasmonic resonances and that the SERS enhancement varies spatially for different analytes. Additionally, we show that the local surface morphology can be estimated by optical methods. Furthermore, it is easy to prepare, uncomplicated, and comparatively cost-effective. Such a substrate enables fast screening of analytes and is stable, with reproducible trends in the results. This approach shows high controllability for the fabrication of such SERS substrates at the micro and nano scales and increases the potential of practical applications of non-uniform SERS substrates.

## 2. Results and Discussion

Several non-uniform samples were prepared as described in the Materials and Methods section; one of them has been illustrated in Figure 1. For ease of explanation, we labeled the substrates as S1, S2, and S3. Figure 1a shows a schematic of such non-uniform SERS substrates (S1, S2, S3), where the center of the evaporation source is marked as “C”. Figure 1b shows SEM images of the points marked P1 to P9 in Figure 1a. The SEM image at P1 shows that the substrate is covered by relatively spherical NPs. On the contrary, the SEM image at P2 shows the formation of connected Au islands forming percolation paths, which gradually increase in size until P5. This inhomogeneity occurs since points P2 to P5 are closer to the center, such that more metal is deposited in this region, and the thermal annealing of the sample leads to the formation of connected Au islands [39].

Afterwards, SEM images at points P6 and P7 show a slow transition from a connected island-like morphology towards particle-like features. Finally, points P8 and P9 of S3 are further away from the center and are covered by relatively spherical particles. Figure 1c shows the particle size distributions at points P1, P8, and P9, as determined with the image processing software ImageJ. At P1, the average particle size (radius) is 15 ± 10 nm, and moving across the island film to P8 leads to larger particles with 20 ± 13 nm. The largest particle density with the smallest particles is observed at P9, with an average particle size of 10 ± 6 nm. The average particle sizes at points P2 to P7 cannot be easily determined since the substrate is covered with connected Au islands. An important parameter for SERS enhancement is the gap between the particles/islands. Figure 1d shows the average gap size between Au NPs and Au islands at points P1 to P9. The gap sizes were evaluated in the image processing software Gwydion. At point P1, the average gap size between the Au NPs is 11 ± 4 nm. As can be seen in the SEM images, at P2, the surface morphology changes from NPs to an island film, which leads to an increase in the gap size to 42 ± 6 nm and 52 ± 5 nm at P2/P3, respectively. Moving further towards the center C of the whole sample at P4/P5, the average gap size between islands gradually decreases to 45 ± 3 nm and 32 ± 3 nm. In this area, more metal is deposited, and the thermal annealing of the sample leads to the formation of larger connected islands with smaller gap sizes. Moving further across the substrate, the trend is reverted, and the average gap size increases again at P6 to P8, and decreases when NPs are formed at P8/P9. Finally, at point P9, the average gap size is 11 ± 3 nm, which is similar to P1. This morphology is similar for different fabricated substrates within the limits of the production process of inhomogeneous samples, as can be seen in the Appendix A.

Figure 2a shows a contour plot of extinction spectra acquired across samples S1, S2, and S3 shown in Figure 1. Spectra were recorded at 40 different locations, as shown schematically in Figure 1a along the substrates (cf. “horizontal position” in Figure 2a,e). The selected extinction spectra in Figure 2b–d correspond to points P1–P9 of the SEM images of S1, S2, and S3 in Figure 1b. The horizontal position locationof P1 to P9 can be seen in Figure 2a.

In general, the extinction maximum of the spectra (illustrated by the black markers in Figure 2a) acquired across the three substrates shifts to longer wavelengths from S1 to S2, and to shorter wavelengths from S2 to S3. Furthermore, the full width at half maximum (FWHM) of the extinction spectra changes from S1 to S3. The extinction spectra of S1 in Figure 2a and the spectra at P1 to P3 in Figure 2d show that the localized surface plasmon resonance (LSPR) is comparably narrow at position P1 and gradually red shifts, broadens, increases in intensity, and becomes more asymmetric towards position P3. This trend continues until the middle of substrate S2, where the spectra are highly asymmetric and are tailing towards larger wavelengths. Moving further to S3, this trend is reverted and the spectral maximum shifts to the blue. Furthermore, the spectral peak sharpens, decreases in intensity, and becomes more symmetric. On substrate S3 the spectral shape is similar to that on S1, and the spectra exhibit a well-defined maximum, indicating a localized plasmon resonance. This alteration of the spectral shape allows us to draw conclusions on the film morphology at the different positions on the substrate. At the beginning/end of S1/S3, the spectra show a clear LSPR resonance, similar to that of a gold sphere [40,41,42]. This is also supported by the SEM images in Figure 1b. However, comparing the spectral position of the experimental extinction maxima with simulations based on Mie’s theory of spherical gold particles (see Appendix A), it is obvious that the experimental resonances show a large red shift. This indicates a significant contribution of coupling between the NPs, which is favorable for SERS experiments, as it creates intense hotspots in the gaps [43]. Moving towards S2, the spectral maximum shifts further to the red, which suggests that the particle size and/or the coupling between the particles increases. The SEM images in Figure 1b show that the film morphology changes from NPs to a connected Au island film towards the center C. Here, more material is deposited due to the evaporation geometry. This morphology change is also visible in the extinction spectra, since they further red shift, increase in intensity, and become strongly asymmetric in the center of the three substrates. In particular, this asymmetry can be a measure of the film morphology, where a large asymmetry indicates an island-like film, and a small one suggests spherical NPs. These effects are summarized in Figure 2e, where the spectral maximum, the amplitude, and the symmetry factor are shown as a function of the horizontal position. The LSPR spectral position and the intensity of the spectra correlate with each other and show similar trends. Moving across the three substrates, the LSPR (shown in black) red shifts from 580 nm to 660 nm and remains approximately constant at 630 nm on S2. For S3, the LSPR blue shifts from 620 nm to 570 nm. The intensity shows a similar trend, and the maximum extinction is observed at the center C. An increase in intensity indicates an increase in optical cross-section due to more material being deposited in those regions and vice versa. This is supported by SEM images in Figure 1b. Additionally, we characterized the asymmetry of the spectra by a symmetry factor, which is the ratio of the difference of the areas on the right and left sides of the spectral maximum normalized to the full area of the spectrum (see Appendix A). A negative symmetry factor means that the area under the curve on the long-wavelength side of the spectral maximum is larger than the area on the shorter-wavelength side. This symmetry factor is shown by the green markers in Figure 2e. In general, the symmetry factor is anticorrelated with the spectral maximum and the intensity. As the LSPR red shifts, the extinction spectrum is increasingly tailing towards larger wavelengths, and thus, the symmetry factor shifts to increasingly negative values. It remains approximately constant for S2 and gradually decreases in amplitude for S3, indicating that the extinction spectra are becoming more symmetric. Hence, a symmetry factor close to zero indicates a particle-like morphology, and a negative symmetry factor indicates a connected island film.

The potential of such a non-uniform substrate for SERS is shown by using 1,2-bis(4-pyridyl) ethylene (BPE) and methylene blue (MB) as analytes. For this purpose, droplets of suspensions of the respective analyte are positioned on the substrate, as shown in the Materials and Methods section below. The analyte BPE was evaluated at 12 different horizontal positions, whereas 20 horizontal positions for MB were analyzed across the substrate. Note: Due to the symmetry of the substrate, the 12 or 20 droplets were placed at distances of 2–3 mm between positions P1 and P5 by hand. The focus spot sizes for the SERS measurements is in the order of few μm, much larger than the size of the individual nanostructures and much smaller than the droplet size.

The most intense peak of BPE, namely the benzene ring stretching (C-H) at 1608 cm^−1^, was chosen for analysis. Raman measurements and corresponding extinction spectra acquired at the same specified points (without analyte) are shown in Figure 3a–c. Please note that different substrates were used for the SEM characterization in Figure 1 and Figure 2 and for the SERS experiments in Figure 3 due to carbon residuals after the SEM measurements.

The extinction spectra in Figure 3a were acquired before deposition of the analytes and show a similar trend as previously observed in Figure 2, suggesting a continuous transition from spherical particles at position 1 to a connected islands film at position 12. The extinction maxima of these spectra are shown in Figure 3c by the black line as a function of the horizontal position. The values are determined using an in-house MATLAB script that also considers the asymmetry of the curve, since in curves 10 to 12, the maxima are no longer discernible. The spectral shape of the extinction spectra suggests that this substrate covers the morphology change between points P1 and P5 of the substrate shown in Figure 1 and Figure 2. For this substrate, there is a red shift from approximately 550 nm at position 1 to 670 nm at position 12. This means that the particle sizes are small at the beginning and gradually increase in size until a connected Au island film is obtained. Note that adding water droplets for the SERS measurements is expected to systematically shift the LSPR resonances to longer wavelengths. The transition from NPs to an island film can again be seen by the broadening and decreasing symmetry of the extinction spectra. The symmetry factor as a function of the horizontal position is plotted in Figure 3c—green line. The spectrum remains symmetric and only broadens gradually until position 3, indicating that the particles in this region just increase in size, but remain roughly spherical. Moving towards position 12, islands are forming and increasing in size. As a consequence, the gap sizes first increase and then decrease again when the islands become larger. Hence, the morphology at positions 1–12 roughly correlates with the SEM images in Figure 1b of P2–P5. Figure 3b shows a series of Raman spectra of BPE at the 12 positions, and the Raman intensity variation as a function of the horizontal position is summarized in Figure 3c—blue line. The most intense peak of BPE (@1608 cm^−1^) is chosen to clearly visualize the Raman enhancement along the 12 lateral positions. However, all other peaks of BPE show the same trend, as can be seen in Figure 3b. The Raman intensity of the 1608 cm^−1^ peak increases from position 1 to 3. Here, the red shift of the extinction spectra suggests that the NP size increases. The maximum Raman enhancement is observed at position 3, where the film morphology is still particle-like with comparably small gaps and the extinction maximum in water is presumably spectrally close to the excitation wavelength (633 nm, black dashed line in Figure 3b). Hence, there is a strong absorption of the excitation light and, as a result, the plasmon excitation is very efficient, and the small gaps are creating intense hotspots, further enhancing the Raman intensity. Afterwards, the Raman intensity gradually decreases from position 4 to position 8. The extinction maximum at these lateral positions red shifts from 605 nm to 625 nm, and the symmetry factor gradually decreases from positions 4 to8. This suggests that the film morphology changes from particle-like to connected islands with larger gaps (see Figure 1). These larger gaps lead to a comparably small Raman enhancement. Hence, this film morphology is not favorable to efficiently enhance the Raman signal of BPE, at least compared to the other morphologies. From position 11 to 12, the extinction maximum further red shifts from 630 nm to 650 nm, and the symmetry factor also becomes more negative, suggesting a further growth of the islands. As previously observed in Figure 1, further growth of the islands leads to a reduction in the gap size, resulting in the small increase in the Raman intensity towards position 12.

To demonstrate that such a non-uniform SERS substrate offers fast screening of optimal SERS conditions for different molecules, we compare the results obtained for BPE with those obtained for MB. A fresh non-uniform SERS substrate was fabricated, and MB was about equidistantly deposited at 20 different lateral positions across the substrate. Raman measurements and corresponding extinction spectra (without analyte) acquired at those points are shown in Figure 3d–f. Again, the extinction spectra in Figure 3d show a trend suggesting a gradual transition from spherical particles at position 1 to a connected islands film at position 20. The extinction maxima of these spectra are shown in Figure 3f by the black line as a function of the horizontal position. Similar to Figure 3a, the extinction spectra suggest that the substrate film morphology in the range of points 1 to 20 is comparable to that between points P1 and P5 of the sample shown in Figure 1 and Figure 2. For this substrate, there is a red shift from approximately 560 nm at position 1 to 660 nm at position 20 and a similar evolution of the spectral shape. This means that the morphology of all three substrates shown here changes in a very similar way under identical substrate preparation. The symmetry factor as a function of the horizontal position is plotted in Figure 3f—green line. The spectrum remains symmetric and only broadens gradually until position 10, depicting that the particles in this region just increase in size, but remain roughly spherical. Moving further across the substrate, islands are forming and increasing in size. As a consequence, the gap sizes first increase and then decrease again when the islands become larger. This covers the morphology change observed in the SEM images in Figure 1b of P1–P5. Figure 3e shows a series of Raman spectra of MB at the 20 positions; the most intense peak of MB (@1635 cm^−1^) is chosen to clearly visualize the Raman enhancement along the 20 lateral positions. However, all other peaks of MB show the same trend, as can be seen in Figure 3e. The Raman intensity variation of the 1635 cm^−1^ peak as a function of the horizontal position is shown in Figure 3f—blue line. The Raman intensity from position 1 to 20 has a wavy nature, which requires further investigation. The maximum Raman enhancement is observed at position 14, where the Au film consists of connected islands with comparably small gaps with an extinction maximum without analyte (610 nm) spectrally close to the excitation wavelength (633 nm, black dashed line in Figure 3d). Hence, there is a strong absorption of the excitation light and, as a result, the plasmon excitation is very efficient and the small gaps are creating intense hotspots, further enhancing the Raman intensity. Afterwards, the Raman intensity gradually decreases from position 14 onwards to position 20.

Since we have used different substrates for BPE and ME, it is important to compare the results for similar substrate morphologies. This is shown in Figure 4 by plotting the SERS intensity against the extinction maximum of the substrate (without analyte).

The highest SERS intensity for BPE (solid blue line in Figure 4) is obtained at position 3, where the maximum extinction of the substrate without analyte is at 585 nm. At this position, the film morphology is particle-like with small gaps. On the other hand, the highest SERS intensity for MB (dashed blue line in Figure 4) is obtained at position 14 with a substrate resonance without analyte at approximately 610 nm. At this position, the film consists of connected islands with small gaps. Hence, these two different molecules show two different laterally separated resonant positions on a non-uniform SERS substrate. This indicates that the type of non-uniform SERS substrate produced for this study offers multiple resonant conditions. Depending on the excitation wavelength and properties of the molecules, the maximum enhancement is observed at different lateral positions on those samples. This is a clear indication that these types of gradient samples can be used to screen for the optimal SERS conditions for different molecules. This also shows that metal islands can be tuned hassle-free and can be used as comparatively low-cost SERS substrates. Furthermore, the parameter range of these substrates can be further expanded by different annealing processes or by patterning of the underlying substrate.

## 3. Materials and Methods

### 3.1. Non-Uniform Metallic Au Particles Substrate Fabrication

The SERS substrates were prepared by metal evaporation in a vacuum evaporator (Plassys MEB400, France) using the thermal evaporation mode. The evaporation rate was 0.03 nm/s monitored by a built-in quartz crystal sensor. The working pressure was 1.0 × 10^−6^ Torr. Prior to metal evaporation, the glass substrates were washed with a mixture of detergent (Decon 90) and ddH_2_O (1:9 volume ratio) in an ultrasonic water bath at 50 °C for 20 min, and then dried within an N_2_ stream. Figure 5a shows a schematic of a conventional metal evaporation technique to deposit layers of uniform thickness covering the whole substrate. Non-uniform SERS substrates were fabricated by evaporating a thin layer of gold (Au), followed by thermal annealing, which leads to the formation of Au NPs/islands. To attain non-uniform deposition of metal NPs, the sample holder, containing bare glass substrates, was placed very close to the source. This arrangement is shown in Figure 5b. The distance between source and substrate in this work was 70 mm. Afterwards, the sample is thermally annealed at 250 °C for 30 s on a hot plate. Figure 5c shows an optical image of the resultant non-uniform Au SERS substrates. The sample holder can hold 3 glass substrates of 22 mm × 40 mm each; the non-uniformity is extended over the 3 glass substrates and is even visible by the naked eye (Figure 5c).

### 3.2. Extinction Spectra

Extinction spectra were recorded in two confocal systems (Setup 1: Figure 2a–c and Figure 3a, Setup 2: Figure 3d). The region of interest in each case is illuminated with a nearly collimated white light source under normal incidence. The transmitted light is collected by an objective, spatially filtered by either an optical fiber or a pinhole, and detected by a spectrometer [44,45]. Due to the spatial filtering, signal from an area of ∼10 μm diameter is recorded. All measurements were conducted at room temperature and ambient conditions. 

Setup 1: Illumination takes place with a halogen light bulb in a LabRAM 800HR equipped with an Olympus BX41 microscope. The transmitted light is collected by a 20× objective (NA 0.45) and coupled into an Ocean Optics QE 65,000 spectrometer through a 200 μm fiber. 

Setup 2: Illumination takes place with a halogen light bulb in a Nikon Ti-U inverted microscope. The transmitted light is collected by a 20× objective (NA 0.5) and spatially filtered by a 200 μm pinhole in the image plane before entering an Ocean Optics QE 65,000 spectrometer. 

Extinction was evaluated in the spectrometer software according to
Extinction = −log_10_ (Ts/T_0_),
where T_0_ represents the light intensity transmitted through a bare glass, and Ts is the intensity transmitted through the fabricated gradient SERS substrates. Both T_0_ and Ts were first corrected for the dark current of the spectrometer’s CCD detector.

### 3.3. SERS Experiments

A LabRAM 800HR from Horiba equipped with an Olympus BX41 microscope, a 632.8 nm laser and a 10× (NA = 0.3) objective in a reflection configuration was used for illumination and collection of the Raman signal. SERS experiments were performed with 1,2-bis(4-pyridyl) ethylene (BPE) with a concentration of 0.25 × 10^−3^ M and methylene blue (MB) with a concentration of 0.25 × 10^−3^ M in water. The excitation and emission maxima of the fluorescent molecule BPE are located at 388 nm and 456 nm, respectively [46,47], whereas the excitation and emission maxima of MB are located at 400 nm and 686 nm, respectively [48]. For the SERS experiments, droplets of 1 μL each were about equidistantly deposited on the substrate (corresponding to droplets with diameters of several 100 μm at a spacing of ~2–3 mm), and spectra were acquired rapidly before the droplets dried. The location of the droplets is denoted as “horizontal position” in the evaluation of the SERS spectra. Drying of the analyte might cause a non-uniform distribution of the analytes over the substrate surface, resulting in accumulation of the analyte at SERS hotspot regions, eventually hindering a correct characterization of the substrate [49,50]. Furthermore, it is worthwhile to validate the SERS substrate in solution to show its usefulness as a molecular sensor. All the measurements were conducted at room temperature and ambient conditions.

### 3.4. Scanning Electron Microscopy (SEM)

SEM images were acquired with a Schottky Jeol 6500F at 2 kV, a secondary electron detector, and an accelerating voltage of 0.5 to 30 kV.

## 4. Conclusions

We have demonstrated a straightforward strategy to design and fabricate non-uniform SERS substrates that offer a wide range of resonance wavelengths for faster screening of analytes. The substrates were fabricated in a metal evaporator by placing the sample holder close to the Au source. The substrate was characterized with SEM and shows a continuous transition from Au NPs to connected Au islands and back at different lateral positions. The intensity maximum of the extinction spectra showed that the resonances of such substrates vary over a wide range from 560 nm to 680 nm. Furthermore, the spectral shape of the extinction spectrum allows us to draw conclusions on the Au film morphology in that region. A comparatively sharp and symmetric spectrum originates from NPs, while a broad and asymmetric spectral shape indicates a connected island morphology. Additionally, the intensity of the spectra increases, since the optical cross-section is increased for islands with a larger area. Hence, acquiring an extinction spectrum can serve as an easy tool to characterize the morphology of an Au film based on the previous knowledge from the SEM images. The SEM images also reveal that the gap size between NPs/islands varies with the amount of material being deposited in that region. Smaller gap sizes between particles and islands create hotspots that increase the Raman enhancement. Such substrates are suitable to finding the optimal surface morphology for SERS. We investigated the SERS efficiency for BPE and MB and found that the maximum SERS enhancement for these molecules occurs at different lateral positions and thus, for different surface morphologies. For the non-resonant molecule BPE, the strongest enhancement was observed when the extinction maximum was close to the excitation laser wavelength, while the film morphology is still particle-like with small gaps was maintained. However, this was not the case for the resonant molecule MB, where it was favorable to have the extinction maximum at an about 20 nm longer resonance wavelength (possibly still closer to the excitation laser wavelength), even though the actual morphology consists of connected islands with larger gaps. (Note that the measured extinction maximawill be shifted to longer wavelengths when the analyte is added.) Hence, such non-uniform SERS substrates provide cost-effective and fast screening of analytes, show reproducible trends, and are reliable and easy to fabricate.

## Figures and Tables

**Figure 1 molecules-27-05097-f001:**
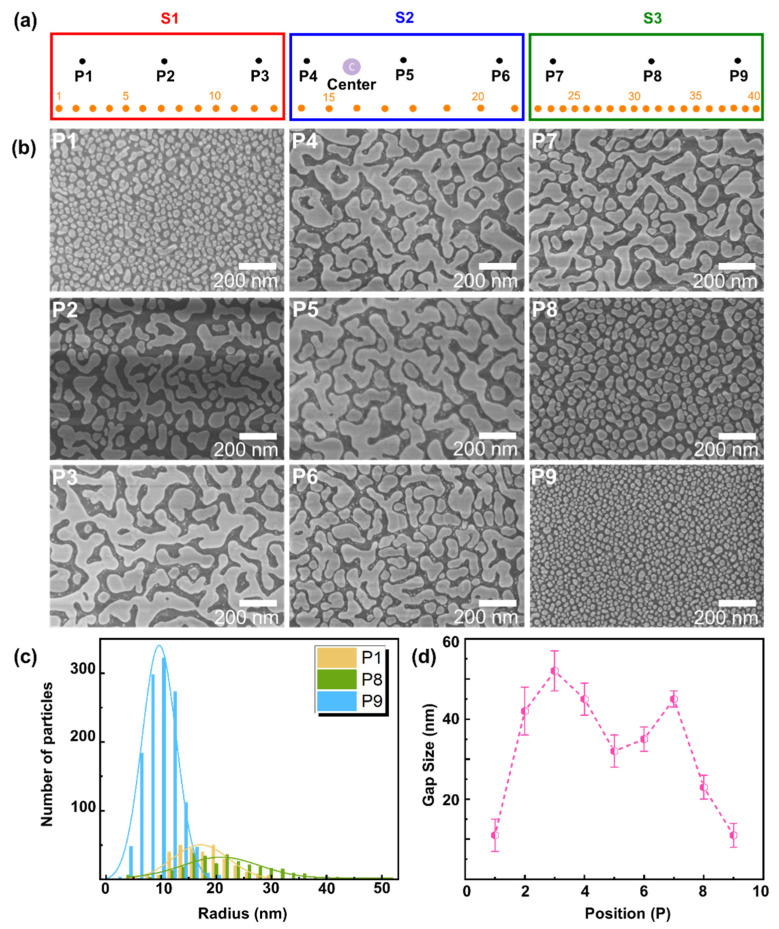
(**a**) Schematic of non-uniform SERS substrates S1, S2, and S3 marked with points P1 to P9 (black, points where SEM images were taken), and with points 1 to 40 (orange, points where extinction spectra were acquired); the length of each substrate is 40 mm. (**b**) SEM images (P1 to P9) at corresponding points marked in (**a**). (**c**) Au particle size distribution at points P1, P8, and P9, and (**d**) average size of gaps between Au NPs/islands at points P1 to P9.

**Figure 2 molecules-27-05097-f002:**
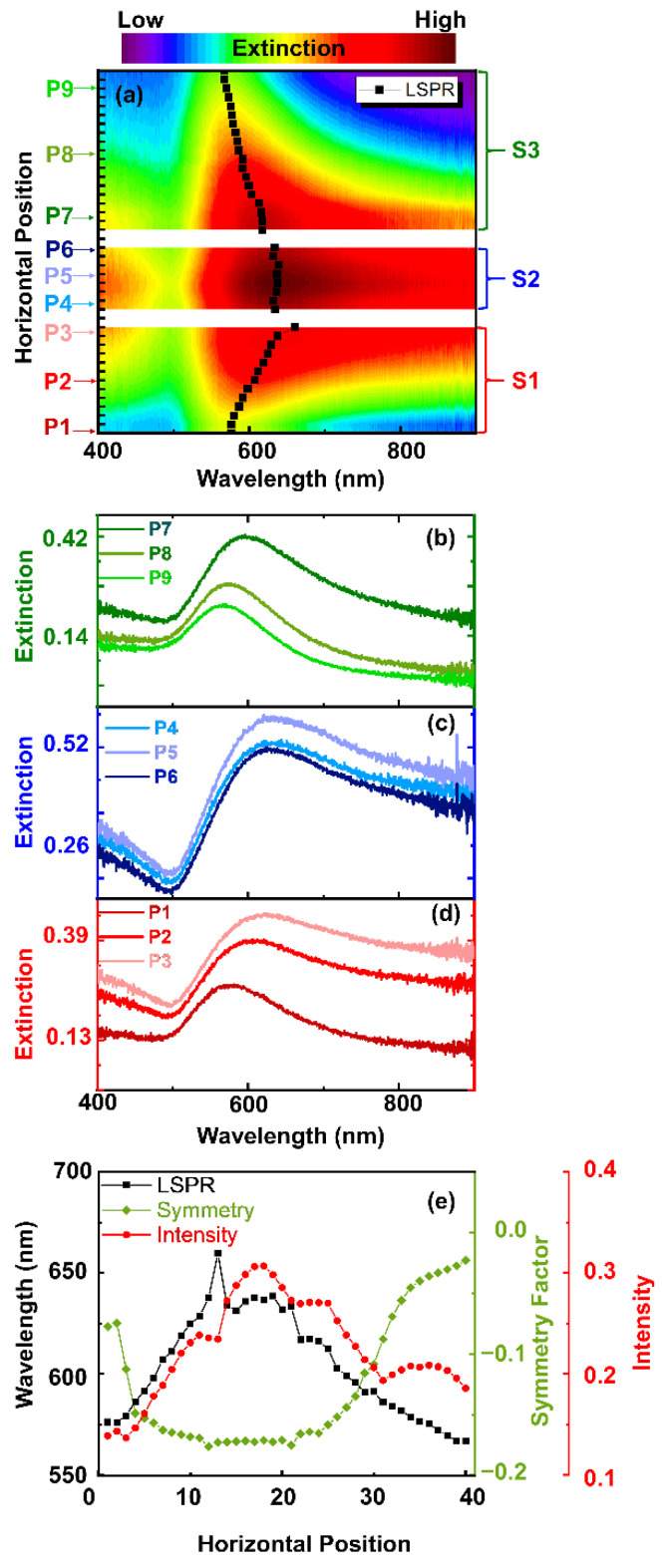
(**a**) Contour of the extinction spectra acquired across substrates S1, S2, and S3, as denoted schematically in orange at 40 locations in Figure 1a. The black markers illustrate the maximum of each spectrum. (**b**) Corresponding extinction spectra of S3 at points P7, P8, and P9, (**c**) corresponding extinction spectra of S2 at positions P4, P5, and P6, (**d**) corresponding extinction spectra of S1 at positions P1, P2, and P3, and (**e**) LSPR, amplitude and symmetry factor of extinction spectra in (**a**).

**Figure 3 molecules-27-05097-f003:**
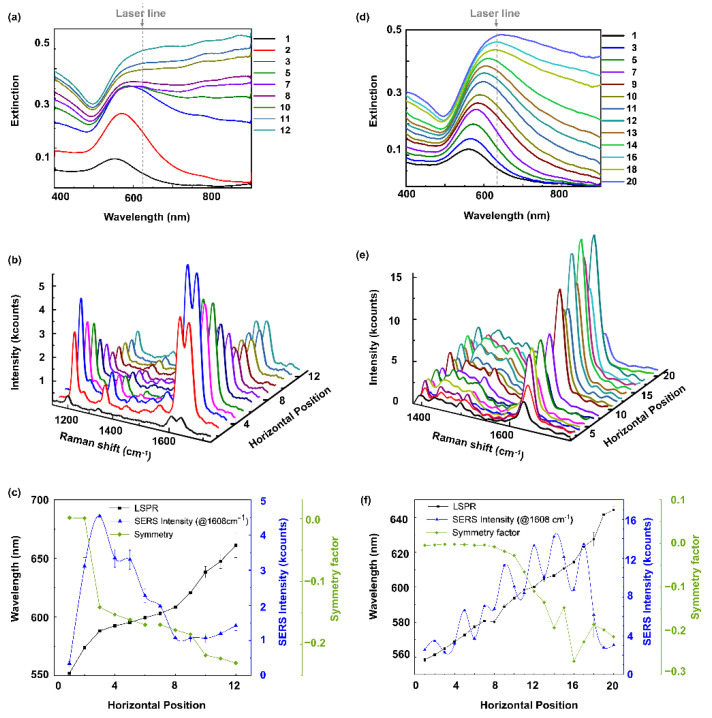
(**a**) Extinction spectra acquired (without analyte) at 12 lateral positions across the substrate used for SERS of BPE. (**b**) Raman spectra of BPE (1 µL—0.25 × 10^−3^ M) acquired at the same 12 positions. (**c**) SERS intensity of BPE (@1608 cm^−1^)—blue curve, spectral maxima of the extinction spectra shown in (**a**)—black curve, symmetry factor of the extinction spectra shown in (**a**)—green curve, (**d**) Extinction spectra acquired (without analyte) at 20 lateral positions across another gradient SERS substrate used for MB. (**e**) Raman spectra of MB (1 µL—0.25 × 10^−3^ M) acquired at the same 20 positions. (**f**) SERS intensity of MB (@1625 cm^−1^)—blue curve, spectral maxima of the extinction spectra shown in (**d**)—black curve, symmetry factor of the extinction spectra shown in (**d**)—green curve.

**Figure 4 molecules-27-05097-f004:**
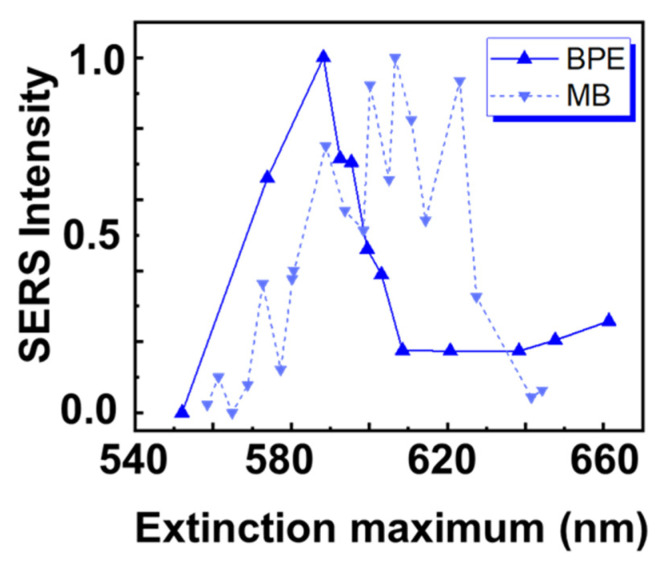
Plot comparing the SERS intensity for the two molecules BPE and MB to the extinction maximum of the LSPR (without analyte).

**Figure 5 molecules-27-05097-f005:**
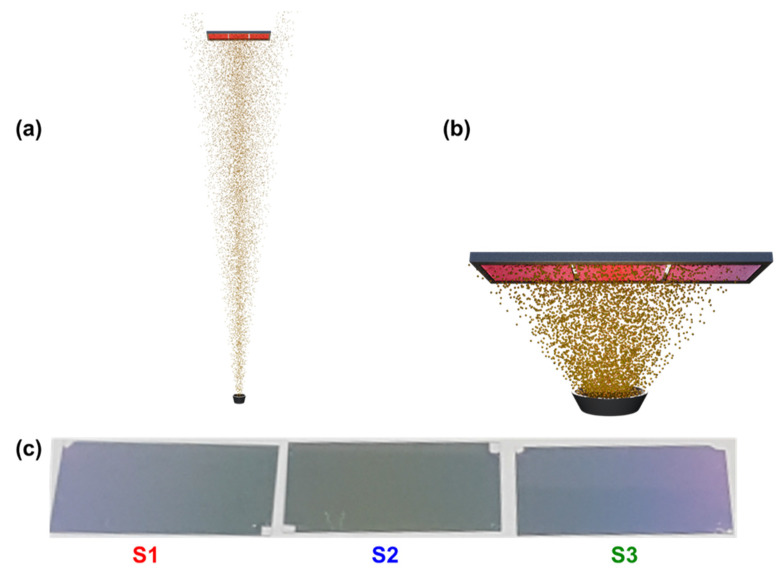
(**a**) Schematic of vacuum evaporator for metal deposition in the conventional method (for approximately uniform thickness of metal deposition), (**b**) schematic of vacuum evaporator for the metal deposition with non-uniform thickness, and (**c**) resulting non-uniform SERS substrates on glass.

## Data Availability

The data sets generated and/or analyzed during the current study are not publicly available since they are part of an ongoing PhD thesis. However, the data sets are available from the corresponding authors on reasonable request.

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
