# Peer review of "Gradient SERS Substrates with Multiple Resonances for Analyte Screening: Fabrication and SERS Applications"

_molecules, 2022, doi:10.3390/molecules27165097_

Round 1

Reviewer 1 Report

Reviewed work titled "Gradient SERS substrates with multiple resonances for analyte 2 screening: Fabrication and SERS applications" provides fundamental information on the design and fabrication of active SERS platforms, where the size and shape of plasmonic metals play a key role. This type of research is important from the application point of view in this area. However, the publication requires a few additions:

1. there is lack of information on what material is the subject of the work, the general information contained in the abstract and the aim of the work is insufficient,

2. introduction: lines 36 and 37, superscript should be used (.... by a factor of 104–106 ........ 108 .......) 

3. results and discussion: the Figures in the publication should be described in the correct order from number 1 to number 5 (.... Several non-uniform samples were prepared as described in the Materials and Methods section, one of them has been illustrated in Figure 1. As shown in Figure 5 (c), the non-uniformity is stretching across the three glass substrates ......)

4. results and discussion: figure 1 - no information about the tested material in the figure's caption

5. does the Authors of the study tried to determine the distances between Au particles / islands by AFM microscopy, the AFM linear profiles could provide a lot of interesting information about the geometry of the obtained samples, taking into account the fact that the metal was deposited on a smooth glass substrate

6. does the Authors of the study tried to make not only linear SERS measurements but also maps of the SERS spectrum distribution for the selected band of the probe molecule, on this basis it could be determined experimentally where the hot spots are located on the surface of the prepared platforms (2D map)

7. results and discussion: correct the numbering of the Figures - twice is the number 3

8. results and discussion: does the „horizontal position” should be scaled in some units of length (is it a microscope table shift relative to the laser beam) - Figures 2 and 3

9. what other non-geometric factors related to the substrate may influence on determined excitation maxima as a result of the LSPR measurements.

Reviewer 2 Report

The paper refers to fabrication and testing of SERS substrates of non-uniform, continuous variable, morphology. While the topic is inciting, and of great interest for Raman spectroscopy applications, there are some issues. The main concern refers to the morphological reproducibility of the substrates, the lack of which would render the technique unusable. While claims of reproducibility are made (21,74,378), there is no convincing supporting data or references provided.

All the comments and observations are listed below.

Testing on analytes, and used analytes are not mentioned in abstract.

Not clear what is the relation between points P1-9, the 40 extinction spectra locations, and positions 1-12 (BPE measurements) and positions 1-20 (MB measurements). For clarity, precise XY coordinates for all locations (deposition center C, P1-9 SEM locations, spectra measurement locations with and without analytes) should be provided, along with schematic of the substrates.

Please, define the acronims LSPR (Localized surface plasmon resonance), FWHM( full width at half maximum) when first used.

Unusual and confusing order of the chapters, placing the "2. Results and discussion" before "3.Materials and methods".  

75 "Such a substrate enables fast screening of analytes and is stable with reproducible trends in the results. This approach shows high controllability for the fabrication of such SERS substrates at the microand nanoscale and increases the potential of practical applications of non-uniform SERS substrates" Please, provide data or reference to support the claims for micro and nanoscale controllability, and reproducibility.

153 "Especially, this asymmetry can be a measure of the film morphology, where a large asymmetry indicates an island-like film and a small one suggests spherical NPs." Please, provide supporting reference.

178 "The analyte BPE was evaluated at 12 different horizontal positions, whereas 20 horizontal positions for MB were analyzed across the substrate." Please, specify the exact positions of the locations.

179 "The spot sizes for the extinction and SERS measurements are on the order of few μm, and therefore much smaller than the droplet size." The extinction spectra were acquired using the substrate with the analyte deposited on the surface? Based on extinction spectra, the similitude in morphology with the SEM investigated substrates is concluded, regardless the presence of analyte?

The extinction spectra presented in fig. 2 b,c,d and fig. 3 a,d seem quite different. Is it due to different locations morphologies, or due to the presence of analytes, or both? Also, the figures cannot be easely compared.

184 "Please note that different substrates were used for the SEM characterization in Figure 1 and 2 and for the SERS experiments in Figure 3 due to carbon residuals after the SEM measurements." Were the substrates of same morphology with the one investigated by SEM microscopy?

201 "For this substrate, there is a red shift from approximately 550 nm at position 1 to 670 nm at position 12. This means that the particle sizes are small at the beginning and gradually increase in size until a connected Au island film is obtained." Confusing, because the exact position in relation with SEM investigated areas is not specified in the first place.   

239 "Again, the extinction spectra in Figure 3 (d) show a trend suggesting a gradual transition from spherical particles at position 1 to a connected islands film at position 20."

242 "Similar to figure 3 (a) the extinction spectra suggest that this substrate is analogous to the points P1-5 of the one shown in Figure 1/2." Very confusing... The positions 1-20 were all between point P1 and point P5?

268 "This is shown in Figure 4 by plotting the SERS intensity against the extinction maximum of the substrate." There is no figure 4.

Figure 3 between lines 269-270: It is very confusing, cannot understand. The comments in text don't clarify.  

292 Specify the gas that was used, please.

297 "Non-uniform SERS substrates were fabricated by evaporating a thin layer (5 nm) of gold (Au)". Specify the geometry and area of the thin layer, please.

300 If possible, provide the distance from vapor source to substrate for non-uniform deposition, please.

332 Typo: A LabRAM 800HR from Horiba equipped which equipped with Olympus BX41 a.
